# Optrode-Assisted Multiparametric Near-Infrared Spectroscopy for the Analysis of Liquids

**DOI:** 10.3390/s24030729

**Published:** 2024-01-23

**Authors:** Maria Giulia Delli Santi, Salvatore Castrignano, Marialuisa Capezzuto, Marco Consales, Patrizio Vaiano, Andrea Cusano, Gianluca Gagliardi, Pietro Malara

**Affiliations:** 1Consiglio Nazionale delle Ricerche, Istituto Nazionale di Ottica (CNR-INO), Via Campi Flegrei 34, 80078 Pozzuoli, Italy; mariagiulia.dellisanti@ino.cnr.it (M.G.D.S.); marialuisa.capezzuto@ino.cnr.it (M.C.); gianluca.gagliardi@ino.cnr.it (G.G.); pietro.malara@ino.cnr.it (P.M.); 2Optoelectronics Group, Engineering Department, University of Sannio, C.so Garibaldi 107, 82100 Benevento, Italy; consales@unisannio.it (M.C.); pvaiano@unisannio.it (P.V.); acusano@unisannio.it (A.C.)

**Keywords:** fiber optics sensors, lab-on-fiber sensors, portable spectrometers, NIR spectroscopy, multiparametric analysis

## Abstract

We demonstrate a sensing scheme for liquid analytes that integrates multiple optical fiber sensors in a near-infrared spectrometer. With a simple optofluidic method, a broadband radiation is encoded in a time-domain interferogram and distributed to different sensing units that interrogate the sample simultaneously; the spectral readout of each unit is extracted from its output signal by a Fourier transform routine. The proposed method allows performing a multiparametric analysis of liquid samples in a compact setup where the radiation source, measurement units, and spectral readout are all integrated in a robust telecom optical fiber. An experimental validation is provided by combining a plasmonic nanostructured fiber probe and a transmission cuvette in the setup and demonstrating the simultaneous measurement of the absorption spectrum and the refractive index of water–methanol solutions.

## 1. Introduction

Near-infrared spectroscopy (NIRS) is a well-established analytical chemistry technique mostly implemented in industrial and agricultural applications as a non-invasive method for quality or authenticity control [1,2,3,4], which is gaining new momentum thanks to the recent acceleration in the miniaturization of related devices [5,6,7,8]. However, NIR spectrochemical analysis, i.e., the identification of spectral features in the sample absorption and their association to specific chemical components, is not straightforward. This wavelength region, as opposed to the mid-infrared, is in fact populated by broad molecular overtone and combination bands, which lead to complex spectra with many superimposed features, in particular, in liquid samples. Also, NIR spectral features are often masked by a complex baseline due to the matrix in which the target components are dissolved. For these reasons, calibrating NIRS spectra against primary standards in simple solutions is almost impossible, and quantitative NIR spectrochemical analysis relies primarily on the use of large calibration sets and chemometric algorithms [9,10]. These algorithms are based on multivariate analysis: they perform a similarity search of the spectral absorption patterns in many-dimensional space. The possibility of assisting the conventional absorption measurement with the quantification of the sample’s physical parameters (like the refractive index, pH, and temperature) allows adding constraints to the NIRS interpretation, increasing its ability to identify an unknown chemical composition.

Optical fiber sensors represent a major resource for this task. Fiber optics and the relative optoelectronic components are indeed easy to integrate in NIR spectrometers, since the two technologies share the same spectral region of operation. Furthermore, fiber sensors can be designed to detect a broad range of physical and chemical properties [11,12]. Optrodes and lab-on-fiber metatips can detect RI, pH, or specific chemicals in a sample thanks to the immobilization coatings or nanostructures fabricated on the fiber surface [13,14]; fiber Bragg gratings (FBGs), traditionally used for strain/pressure measurements, can be equipped with specific polymer coatings that swell in response to chemical composition, moisture, or temperature variations, transducing these parameters to a detectable mechanical stress [15,16,17]. In general, most fiber optic sensors are based on a photonic or a plasmonic resonance whose central wavelength shifts in response to variations in the target parameter [18,19]. This design makes their response independent of the intensity fluctuations of the interrogating radiation; but, on the other hand, it requires a spectral-domain readout.

In this work, we propose an innovative measurement scheme that integrates fiber optic sensors in an NIR spectrometer, allowing measuring in parallel the absorption spectrum of a homogeneous liquid sample and the spectral response of one or more fiber optic probes immersed in it. The technique is based on an evaporating-droplet fiber interferometer [20,21] that encodes the spectrum of an input radiation in a modulation of the fiber-guided intensity. The droplet interferogram is divided by a fiber splitter and used to interrogate the sample with multiple parallel sensing units. The spectral readouts of these sensors are obtained from their output interferograms with an FT-based routine. 

We experimentally demonstrate the method described by using an absorption cuvette and a plasmonic fiber-optic metatip to analyze the broad NIR absorption spectrum and the RI of a liquid analyte, at the same time. The capability of parallel spectral readout demonstrated for these two devices is directly scalable to an arbitrary number of sensing units, thus making the presented scheme the basic structure of a modular all-fiber optic multiparametric NIR spectrometer.

## 2. Materials and Methods

The described measurement scheme is sketched in Figure 1: a supercontinuum radiation source (SC, 1000–2000 nm) is coupled to the broadband fiber splitter S1 and directed towards a thermally expanded core fiber-end encased in a FC/PC ceramic ferrule, where a Perfluorohexane (C_6_F_14_) droplet is deposited. The input radiation is reflected back into the fiber by the fixed fiber–liquid interface and by the droplet upper surface. During the droplet evaporation, the latter surface recedes, and these reflections generate an interferogram that back-propagates towards S1 [20,21]. 

The system effectively behaves as a scanning interferometer with an instantaneous pathlength difference (DL) equal to twice the droplet optical thickness. The 1 × 3 splitter S2 divides the intensity of the droplet interferogram and delivers it to the spectrometer’s sensors: a 2 mm quartz transmission cuvette and a fiber-optic plasmonic probe. The photodetectors Det1 and Det2 record the cuvette transmission *I_T_*(*t*) and the probe reflection *I_R_*(*t*). The third branch of the splitter S2 brings the signal to Det3 and to Det4. Det3 records the unaltered droplet interferogram *I*_0_(*t*), while Det4 records its *λ_ref_* = 1550 nm component *I_ref_*(*t*), isolated by the coupler S3 and a fiber notch filter. 

To obtain the spectral readout of the deployed sensors via Fourier transform, the recorded time-domain interferograms must be first converted into functions of the interferometer’s pathlength difference Δ*L*. The single-wavelength interferogram *I_ref_*(*t*) serves this purpose: in this signal, the time interval of each fringe corresponds to a fixed variation of the interferometer pathlength difference of 2*nλ_ref_*, with *n* being the refractive index of the droplet fluid. Since all the recorded signals are synchronous (they are generated by the same droplet evaporation), equal time intervals correspond to equal pathlength variations, which allows transposing *I_T_*, *I_R_*, and *I*_0_ in Δ*L* units. 

The interferograms so processed are finally Fourier transformed to return *I_T_*(*λ*), *I_R_*(*λ*), and *I*_0_(*λ*) that yield the normalized NIR absorption spectrum and the profile of the probe’s plasmon resonance (see Figure 1). A more detailed description of the described analysis procedure can be found in ref. [20] and its Supplementary Materials.

The fiber-optic probe consists of a lab-on-fiber nanostructure directly fabricated on the tip of a standard single mode optical fiber, accomplishing a plasmonic metasurface formed by a matrix of rectangular nanoholes obtained by means of Focused Ion Beam milling within a 30 nm thick gold substrate [22,23]. Adhesion of the gold substrate to the fiber silica material is granted by a thin (~2 nm) titanium intermediate layer, shown in Figure 2a. The metasurface is based on the unit cell illustrated in Figure 2c, comprising two rectangular apertures (*Lx* = 330 nm, *Ly* = 150 nm) with an orientation of ±45° at a distance *d* = 511 nm from each other, repeated with a periodicity of 2*d* to cover a patterned area of 14 µm × 14 µm aligned to the fiber core, as shown in the scanning electron microscope (SEM) images reported in Figure 2b. The periodic structure supports a 100 nm wide localized surface plasmon resonance (LSPR) in P polarization that shifts toward longer wavelengths with the increasing RI of the surrounding medium in which the probe is immersed. A fiber polarizer at the probe input is used to maximize the coupling to the plasmon resonance, while the reflected radiation, representing the probe’s output, is directed to Det2 by a fiber optic circulator. The operating bandwidth of the circulator limits the span of the calculated spectra (see example in Figure 1), but it is still sufficient to record the entire LSPR profile. The LSPR response to RI changes was calibrated with sucrose solutions at known concentrations and corresponds to a sensitivity of about 300 nm/RIU, in line with the average sensitivity of fiber-optic plasmonic sensors [12].

## 3. Results

The described setup was tested by analyzing methanol–water solutions up to 50 percent water weight concentration. At each measurement, a Perfluorohexane droplet was placed on the fiber tip. The interferograms generated at the various interferometer arms during its rapid evaporation were recorded and processed as described in the previous section to obtain the normalized absorbance spectrum of the methanol solution *I_T_*(*λ*)/*I*_0_(*λ*) and the plasmon resonance profile of the immersed probe *I_R_*(*λ*)/*I*_0_(*λ*). 

The absorbance and LSPR spectra obtained with the various solutions are shown in Figure 3a,c, with each color corresponding to a different concentration (indicated in panels b and d). Thanks to the broad spectral coverage of the SC radiation and the wideband operation of the fluorocarbon droplet interferometer [21], the recorded spectra span from 1200 to 2000 nm. Their resolution (~1 nm), unnecessarily high for our liquid samples, allows smoothing the curves by a 10-point adjacent average without losing information.

## 4. Discussion

In the absorption spectra of Figure 3a, the NIR pattern of methanol is visible (black line), with the distinctive OH and CH overtones @1450 and 1700 nm. As water is added to the solution (colored profiles), the absorbance of the OH band @1450 nm increases, and another stronger band @1950 nm emerges (OH stretch/deformation combination). Figure 3b shows the absorbance of the OH water bands, calculated by subtracting the methanol contribution (black line) from the spectra and integrating the percent absorption over their entire width. For the 1950 nm band, which is not entirely captured, twice the half integral was considered. Moreover, Figure 3b reveals a linear absorbance trend for the weaker 1450 nm band (showing signs of saturation for concentration above 30%) and a strongly saturated trend for the absorption of the 1950 nm band.

Figure 3c,d display, respectively, the LSPR reflectance profiles (zoomed around the resonance center for ease of visualization) and their central wavelength as a function of the water concentration. In pure deionized water and pure anhydrous methanol, the LSPR sets at 1499 nm and 1527 nm (dashed and black profiles in Figure 3c). Since water and methanol have an almost identical RI (*n_water_* = 1.3164 and *n_methanol_* = 1.3174 [24]), such a large shift indicates an LSPR responsivity of about 24,000 nm/RIU, about 80 times larger than the response measured with sucrose solutions. This surprisingly high sensitivity allows, in fact, resolving methanol dilution steps as small as a few percent. A refractive index scale extrapolated assuming the linearity of the LSPR response is shown on the right-hand side of Figure 3d. 

Also, the LSPR shifts towards smaller wavelengths as the methanol is diluted, i.e., the refractive index of the solution decreases, in contrast with the increasing trend commonly measured at visible wavelengths [25,26,27]. Unfortunately, the scientific literature completely lacks water–methanol RI measurements in the near infrared region to confirm this behavior. The described RI trend has been reproduced also with a second identical LSPR fiber optic sensor and with a commercial spectrum analyzer, to rule out the possibilities of sensor malfunction or spectral readout artifacts. 

A hypothesis that could explain both the enhanced LSPR response and the decreasing RI trend of our measurements is the occurrence of metal-catalyzed methanol oxidation on the sensor surface, an effect already reported in other Au-Ti plasmonic systems [28,29,30]. In such a scenario, large LSPR shifts are caused by the presence of methanol-derived chemical species with a high refractive index (e.g., formaldehyde, dimethyl ether). Also, the concentration of methanol byproducts in the vicinity of the sensor surface should reduce with increasing dilution, thus reducing the overall RI and producing LSPR shifts towards the blue wavelengths, as those shown in Figure 3d. A detailed investigation on a possible methanol oxidation mechanism on the surface of our fiber probes is not feasible with the current spectroscopic setup, which can only provide the bulk sample absorption. On the other hand, confirming such a mechanism could have an enormous potential for ultrasensitive refractometric detection of methanol, for example in alcoholic beverages. Hyperspectral imaging of the immersed fiber tip [31] could be one viable method for this inquiry, which will be the object of a dedicated work.

## 5. Conclusions

In this work, we have demonstrated an NIR spectroscopic technique based on the parallel interrogation of a liquid sample in a transmission cuvette and with a fiber optic sensor. The absorption spectrum and refractive index, representing in our demonstration an example of characterizing parameters with a spectral readout, are measured in water–methanol solutions simultaneously by a single device. The measurement is realized in an extremely compact and robust setup where the radiation source, the sensors, and the spectral readout unit are all integrated in a telecom optical fiber. The presented system can be readily deployed with an arbitrary number of parallel probes (the only practical constraint to this number being the intensity of the radiation directed to each probe), thus demonstrating a strong potential for the multi-parametric spectrochemical analysis of liquids.

## Figures and Tables

**Figure 1 sensors-24-00729-f001:**
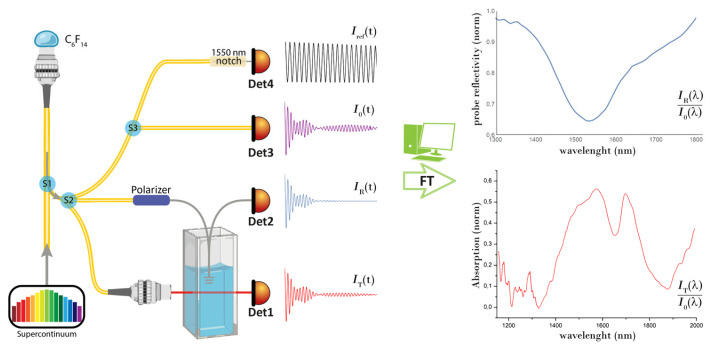
Setup of the fiber-optic multiparametric analyzer. An evaporating fluorocarbon droplet, injected with radiation from a supercontinuum source, backreflects a time-domain interferogram, whose intensity is divided and distributed throughout the system by the splitters S1, S2, and S3. Det1 and Det2 record the analyte-transmitted interferogram *I_T_*(*t*) and that reflected by a fiber optic plasmonic probe immersed in it, *I_R_*(*t*). Det3 and Det4 record the unchanged interferogram *I*_0_(*t*) and a narrow spectral slice extracted from it for normalization and synchronization purposes *I_ref_*(*t*). Subsequent processing and Fourier Transformation (FT) of the recorded interferograms return the spectral readout of the deployed sensors, as shown in the examples on the right hand side of the figure.

**Figure 2 sensors-24-00729-f002:**
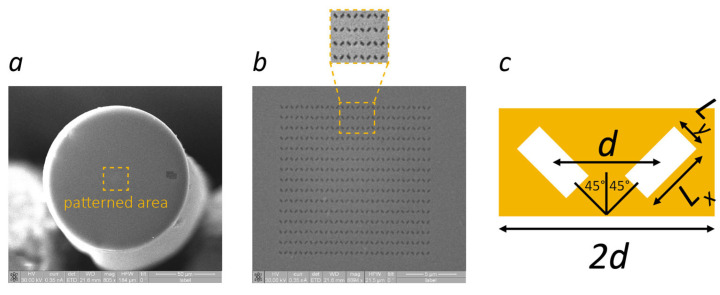
SEM images of the optical fiber hosting the plasmonic metasurface; (**a**) fiber tip and (**b**) a zoom of the patterned area that highlights the periodic repetition of the metasurface unit cell; (**c**) schematic of the unit cell.

**Figure 3 sensors-24-00729-f003:**
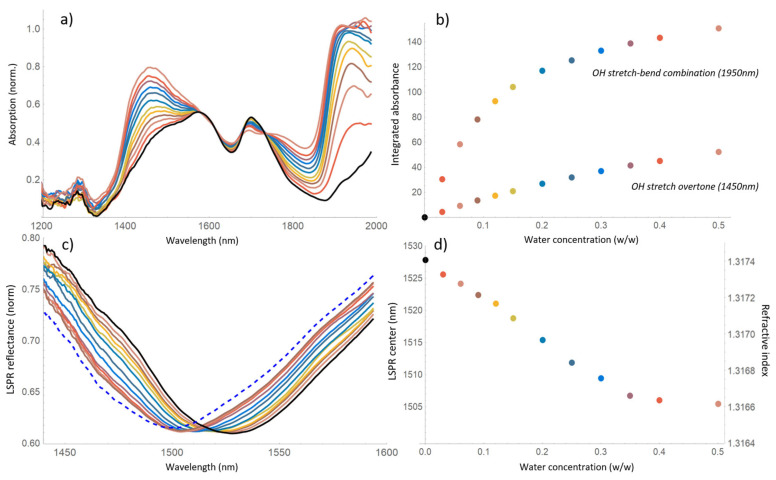
Colors correspond to different water in methanol concentrations in the 0–50% *w*/*w* range, pure methanol: black line (**a**) Absorption spectra of water–methanol solutions, 10 pt adjacent average; (**b**) integrated absorbance of the water bands @1450 and 1950 nm, extracted from the spectra of panel (**a**). (**c**) Localized surface plasmon resonance spectra from the reflectance of the immersed fiber probe, dashed line: pure water, 10 pt adjacent average. (**d**) Resonance positions extracted from the spectra of panel (**c**). Right hand scale of panel d shows the RI of the solutions assuming a linear LSPR response.

## Data Availability

The data presented in this study are available on request.

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
