# Peer review of "Optrode-Assisted Multiparametric Near-Infrared Spectroscopy for the Analysis of Liquids"

_sensors, 2024, doi:10.3390/s24030729_

Round 1

Reviewer 1 Report

Comments and Suggestions for Authors

In this work, the authors have presented a NIR spectroscopic technique based on the parallel interrogation of a liquid sample in a transmission cuvette using multiple fiber sensors. The above paper contains the necessary background introduction, instrument design, and experimental verification. However, some specific problems should be improved to provide more convincing and clearer results for readers. Please see the attached file.

Author Response

Thanks for the comments. In the following, we address them one by one and point to the relative amendments in the manuscript.

  1. It seems that the measurement system is not fully described. I suggest that the authors should explain more about the materials and methods. For example, how did the authors collect the signals from the four photodetectors and process them, using a computer or a microcontroller unit? How did the authors realize the simultaneous control and processing of the four photodetectors? Accordingly, Figure 1(a) is also suggested to be improved for a complete description of the measurement system.

The 4 interferograms recorded by the photodetectors are processed offline, and they are automatically synchronous since they are all generated by the same evaporation process. We edited the “materials and methods'' section (lines 80-101) adding a more comprehensive description of the measurement and the data analysis procedure. With the same purpose, we also rephrased the initial part of the “results” section (lines 141-145), and changed fig.1, including in the illustration the step of transforming interferograms into spectra. Overall, the manuscript has been amended in order to be self-consistent to the reader, but still intuitive. For a  rigorous treatment of the signal processing technique we refer the reader to ref [20] and its supplementary materials.

  1. The authors have stated in the text “The metasurface is based on the unit cell illustrated in Figure 1(b), comprising two rectangular apertures (Lx = 330 nm, Ly = 150 nm) with an orientation of ±45°, repeated with periodicity d = 511 nm to cover a patterned area of 14 μm × 14 μm aligned to the fiber core, as shown in the scanning electron microscope (SEM) images reported in Figure 1(c) and (d).” (line 93 to line 97). Although the authors have marked d in Figure 1(b), it is very hard to determine the position and distance. Therefore, I suggest that the authors should supplement the definition of d and improve Figure 1(b). Additionally, it is also very hard to recognize the ±45° tilt of the rectangular apertures in Figure 1(d). Therefore, I suggest that the authors should improve the resolution of the figure or add a partial enlarged image of two rectangular apertures.

Following the reviewer's comment, we modified the illustration of the metasurface structure (now fig. 2) to improve its readability and to clarify the interpretation of the geometries. The unit cell is a rectangle of sides 2d x d with two rectangular apertures, having sides of lengths Lx and Ly, rotated by + and -45° from the vertical axis, respectively; they are placed vertically in the center, and spaced so that a distance of d exists between their centers. In addition, we included an enlarged image of a portion of the patterned area, from which the alternating orientation of the nanoantennas can be appreciated. Unfortunately, the resolution of the SEM images can’t be improved beyond their original value.  

  1. Although the authors have provided qualitative signals of the four photodetectors, I suggest that the authors should supplement quantitative signals to present a more convincing result.

The raw data of a droplet interferometer and their processing have been described in full detail elsewhere (see answer to comment 1), and we revised the text to explicitly refer the reader to the works where the technique is demonstrated rigorously. In this brief communication we aim at validating the technique for the simultaneous readout of absorption spectra and fiber optic sensors. We thus opted for showing only the spectra to communicate more directly the results and stress their novelty.

  1. Reference 21 was not cited in the main text. Therefore, I suggest a comprehensive check of the references.

We have double-checked the reference section, removing citations that were redundant or too generic (for example former refs 14 and 21) and replacing them with more pertinent references.

  1. It seems that Figure 2(a) is cut off. The authors should provide a complete image.

Fig.2 (now fig.3), has been fixed.

Reviewer 2 Report

Comments and Suggestions for Authors

The manuscript presents study on Optrode-assisted multiparametric Near-Infrared spectroscopy for the analysis of liquids. The presented system can demonstrate a strong potential for the multi-parametric spectrochemical analysis of liquids. Overall, the topic is interesting, and the manuscript got some reliable data from experiments, and the structure of the manuscript is reasonable, the logic is clear, and well written. I suggest that the manuscript be published after minor revision. I do not recommend that authors cite more than 3 articles in the one place, especially in communication. The authors did cite more than four articles in the manuscript (line 31, line 32). Figure 2 needs to improve the quality, which is currently too blurry to read.

Author Response

Thanks for the comments. In the following, we address them one by one and point to the relative amendments in the manuscript.

  1. I suggest that the manuscript be published after minor revision. I do not recommend that authors cite more than 3 articles in the one place, especially in communication. The authors did cite more than four articles in the manuscript (line 31, line 32). Figure 2 needs to improve the quality, which is currently too blurry to read.

We fixed the quality of fig.2 (now fig.3).  We also revised the bibliography, removing unnecessary or redundant citations (for example refs 14 and 21). Large reference groups are sometimes important, especially in the introduction, to give a complete picture variety of a research field. Following the reviewer suggestion, we broke some of them in smaller “topical” clusters.

Reviewer 3 Report

Comments and Suggestions for Authors

The apparatus presented by the authors is original but I find the description rather incomplete. The demonstration example is not enough to show the validity of the method.

What is needed?

-a better description of the method

-a better description of the raw data obtained

-a better description of the calculations needed

-a demonstration of the practical use of the method

Comments on the Quality of English Language

Very minor language weaknesses

Author Response

Thanks for the comments. In the following, we address them one by one and point to the relative amendments in the manuscript.

  1. The apparatus presented by the authors is original but I find the description rather incomplete. The demonstration example is not enough to show the validity of the method

We agree that our demonstrative setup is limited, as it only features two sensing units (absorption cell and a fiber optic sensor). Still, once demonstrated the simultaneous spectral readout of 2 sensing units, the scalability to many is straightforward. In our setup we use the 1X3 all-band bidirectional fiber splitter (S2 in fig.1) to distribute the interferogram to the reference arms and the two sensing units. By using a 1X4 splitter, one more sensor can be readily added. With a 1X5 splitter two more sensors and so on. For any number of sensors added, their output interferograms are synchronous with the det4 single-wavelength interferogram, because they stem from the same droplet interferogram. This allows to convert them in units of interfermeter differential arm-length and thus to Fourier-transform them to get their spectral readouts. This concepts are now explicitly written in the introduction (lines 69-72), in the “materials and methods” section (lines 90-97)  and in the conclusions, where we also mention that the maximum number of devices readable in parallel only depends on how much the droplet interferogram can be fractionated and still remain detectable (lines 211-214).

  1. a better description of the method; 

The “materials and method” section has been extended to better describe the working principle and the measurement methods of our setup and to make it more self-consistent. Anyways, references [20,21] contain additional technical details of the method, which could not be repeated in this work because of the limited space offered by a Communication. The interested reader is explictly referred to these works (line 100-101).

  1. a better description of the raw data obtained -a better description of the calculations needed

The raw data of a droplet interferometer and their processing have already been shown and described in full detail elsewhere. We thus revised the text to explicitly refer the reader to ref [20] and its supplementary materials for these details. In this brief communication, we opted for showing only the spectra in order to communicate more directly the results and stress their novelty.

  1. a demonstration of the practical use of the method.

After this proof of principle experiment, we intend to set up a fiber spectrometer assisted by a larger number of fiber optic sensors, which will be better suited for a practical, real world application.

Reviewer 4 Report

Comments and Suggestions for Authors

This study demonstrates an all-fiber NIR (Near-Infrared) spectrometer capable of interrogating a liquid sample with multiple parallel sensors. The system presented can be deployed with an arbitrary number of parallel probes, showcasing a strong potential for multi-parametric spectrochemical analysis of liquids.

Comment 1:A more detailed description of the experimental setup, especially regarding the integration of fiber optic sensors with the all-fiber NIR spectrometer, would be valuable. Clarification on how these components are synergized and the specific advantages of this configuration would enhance the manuscript's technical depth.

Comment 2:The hypothesis concerning methanol oxidation on the sensor surface is intriguing and warrants further elaboration. Could additional experimental data or theoretical analysis be provided to support this hypothesis?

Comment 3:Considering the advancements in hyperspectral structural equipment, which offer improved spectral resolution and faster imaging speeds, it would be beneficial for this study to discuss the potential of hyperspectral systems in the context of your research.For example, the following article illustrates the use of grating to acquire hyperspectral images, a technique that could enrich your study's methodology.

[1] A Stare-Down Video-Rate High-Throughput Hyperspectral Imaging System and Its Applications in Biological Sample Sensing," in IEEE Sensors Journal, vol. 23, no. 19, pp. 23629-23637, 1 Oct.1, 2023, doi: 10.1109/JSEN.2023.3308394.

Comment 4:The manuscript would benefit from a clearer explanation of the data analysis method, particularly the Fourier transform routine used for spectral readouts. How does this approach ensure the accuracy and reliability of the results?

Comment 5:It is recommended to carefully review the manuscript for formatting consistency, particularly on lines such as 81. Ensure uniform spacing between numbers and letters throughout the text.

Comment 6:The images embedded in the manuscript, like Figure 2, exhibit low resolution, affecting clarity. It is advisable to replace these with high-resolution images for enhanced legibility and effectiveness.

Author Response

Thanks for the comments. In the following, we address them one by one and point to the relative amendments in the manuscript.

  1. A more detailed description of the experimental setup, especially regarding the integration of fiber optic sensors with the all-fiber NIR spectrometer, would be valuable. Clarification on how these components are synergized and the specific advantages of this configuration would enhance the manuscript's technical depth.

We have expanded the description of our experimental setup (lines 80-101) changed fig.1 and added fig.2. Also, both in fig1 and in the text, we have reported how the different fiber components are integrated, in particular, we explicitly mention the presence of the broadband splitters S1, S2, S3 and of the fiber optic circulator used to detect the reflection of the plasmonic fiber probe. The advantage of this configuration is the possibility to integrate near infrared spectroscopy and fiber optic sensing technology in a single all-fiber platform capable of multiparametric measurements in the spectral domain. In order to highlight such claim we rephrased the abstract and part of the introduction (lines 59-72).

2. The hypothesis concerning methanol oxidation on the sensor surface is intriguing and warrants further elaboration. Could additional experimental data or theoretical analysis be provided to support this hypothesis?

To test this hypothesis, a dedicated experiment is necessary. In fact, if methanol oxidation occurs only in the proximity of the sensor surface, a local spectroscopic method is needed to detect it. Unfortunately, our spectrometer is only able to record the bulk absorption of the sample, and can’t provide any conclusive information in this sense. We now mention this explicitly in the conclusions (lines 198-202).

3. Considering the advancements in hyperspectral structural equipment, which offer improved spectral resolution and faster imaging speeds, it would be beneficial for this study to discuss the potential of hyperspectral systems in the context of your research. For example, the following article illustrates the use of grating to acquire hyperspectral images, a technique that could enrich your study's methodology.

The described method could be in principle applied to hyperspectral imaging: the radiation that interrogates the cuvette should be expanded, sent through an objective and recollected with a NIR camera. That would indeed allow to record an interferogram for each pixel and reconstruct a 2D hyperspectral image of the cuvette content. This would allow to resolve a spatially inhomogeneous sample. However, with inhomogeneous samples it wouldn’t make sense to use fiber optic probes, that can only sense a single point in space. In other words, the combination of sensors proposed here only applies to homogeneous fluids, where hyperspectral imaging does not really add significant information. We now state this explicitly in line 61.

4. The manuscript would benefit from a clearer explanation of the data analysis method, particularly the Fourier transform routine used for spectral readouts. How does this approach ensure the accuracy and reliability of the results?

We expanded the “materials and methods'' section (lines 81-102) adding a more comprehensive description of the measurement and the data analysis procedure. With the same purpose, we also rephrased the initial part of the “results'' section (lines 142-146). For a detailed technical explanation and for the demonstration of the accuracy of this method we refer the reader to refs [20] and [21].  

5. It is recommended to carefully review the manuscript for formatting consistency, particularly on lines such as 81. Ensure uniform spacing between numbers and letters throughout the text.

Checked

6. The images embedded in the manuscript, like Figure 2, exhibit low resolution, affecting clarity. It is advisable to replace these with high-resolution images for enhanced legibility and effectiveness.

Checked and revised 

Round 2

Reviewer 1 Report

Comments and Suggestions for Authors

The comments are answered in a suitable manner to achieve the requirement for publishing. However, some minor revisions are also needed.

1. The authors have stated in the text “We experimentally demonstrate the described by using an absorption cuvette” (line 67). Did the authors omit a word between the words “described” and “by”?

2. Why are the displayed wavelength ranges of Probe reflectivity (norm) and Absorption (norm) different in Figure 1? I suggest for the authors explain more about this or improve Figure 1. 

Author Response

  1. The authors have stated in the text “We experimentally demonstrate the described by using an absorption cuvette” (line 67). Did the authors omit a word between the words “described” and “by”?

Thanks, we amended it

  1. Why are the displayed wavelength ranges of Probe reflectivity (norm) and Absorption (norm) different in Figure 1? I suggest for the authors explain more about this or improve Figure 1. 

In each branch, the span of our spectral analysis is determined by the wavelength components of the interrogating radiation that reach the relative detector. In the cuvette branch the transmitted intensity directly impinges the detector, nearly all the wavelengths of the supercontinuum source are recorded and the absorption spectra span from 1200 to about 2000nm.

In the LSPR branch instead the probe reflection is delivered to the detector by a fiber-optic circulator which operates in the 1300-1800nm bandwidth. For this reason the LSPR profile can be reconstructed only in this region. Fortunately, in all our measurements the LSPR resonance is always within this interval. This fact is now explicitly stated in the manuscript (lines 129-130).

Reviewer 3 Report

Comments and Suggestions for Authors

The authors made improvements where possible

The article was improved

Author Response

We thank the reviewer for the comments

Reviewer 4 Report

Comments and Suggestions for Authors

The author has responded to some questions, but there are still some questions that have not been answered.

Comment 1:Does the vertical axis reflection in Figure 3(a) exceed 1?

Comment 2:The author replied to:To test this hypothesis, a dedicated experiment is necessary. In fact, if methanol oxidation occurs only in the proximity of the sensor surface, a local spectroscopic method is needed to detect it. Unfortunately, our spectrometer is only able to record the bulk absorption of the sample, and can’t provide any conclusive information in this sense. We now mention this explicitly in the conclusions (lines 198-202).

Regarding the study of methanol oxidation on sensor surfaces, considering the equipment limitations you mentioned, I suggest exploring the use of hyperspectral imaging technology in future research. This technology can provide more detailed surface reaction information, which helps to gain a deeper understanding of the methanol oxidation process.This is a discussion, and we look forward to the opportunity to conduct this experiment in the future.

Comment 3:The structure and function of hyperspectral imaging equipment are key components that together drive technological progress.Hyperspectral imaging can obtain more local details, which is beneficial for spectral detection of any region of interest in the local area. It would be beneficial for this study to discuss the potential of hyperspectral systems in the context of your research.

[1] A Stare-Down Video-Rate High-Throughput Hyperspectral Imaging System and Its Applications in Biological Sample Sensing," in IEEE Sensors Journal, vol. 23, no. 19, pp. 23629-23637, 1 Oct.1, 2023, doi: 10.1109/JSEN.2023.3308394.

Comment 4:The resolution in some images is still not enough, for example, the text in Figure 2 appears blurry and has some distortion?

Author Response

  1. Does the vertical axis reflection in Figure 3(a) exceed 1?

The normalized absorption by definition cannot exceed 1. In the illustration we slightly extended the scale to accommodate the noise fluctuations of all the spectra around 2000nm. At the edge of our sensing interval, the intensity of the supercontinuum radiation is very small, and when the water concentration is large the OH-band @1950nm absorbs nearly all of it. As a consequence, the signal-to-noise ratio of the detected interferograms is smaller, and the extracted spectral information in this region shows some fluctuations.

  1. The author replied to:To test this hypothesis, a dedicated experiment is necessary. In fact, if methanol oxidation occurs only in the proximity of the sensor surface, a local spectroscopic method is needed to detect it. Unfortunately, our spectrometer is only able to record the bulk absorption of the sample, and can’t provide any conclusive information in this sense. We now mention this explicitly in the conclusions (lines 198-202).
  2. The structure and function of hyperspectral imaging equipment are key components that together drive technological progress.Hyperspectral imaging can obtain more local details, which is beneficial for spectral detection of any region of interest in the local area. It would be beneficial for this study to discuss the potential of hyperspectral systems in the context of your research.

Using hyperspectral imaging as a means to further investigate the local oxidation hypothesis is indeed a good idea. We now specifically refer to it in the text (lines 200-204) and reference the suggested citation (reference [31] in the manuscript).

[1] A Stare-Down Video-Rate High-Throughput Hyperspectral Imaging System and Its Applications in Biological Sample Sensing," in IEEE Sensors Journal, vol. 23, no. 19, pp. 23629-23637, 1 Oct.1, 2023, doi: 10.1109/JSEN.2023.3308394.

  1. The resolution in some images is still not enough, for example, the text in Figure 2 appears blurry and has some distortion?

Figure 2 has been substituted with a higher resolution version of it.

Round 3

Reviewer 4 Report

Comments and Suggestions for Authors

The authors have fully responded to my concerns and recommends publishing this article